# Intranasal delivery of pro-resolving lipid mediators rescues memory and gamma oscillation impairment in $App^{NL-G-F/NL-G-F}$ mice

Ceren Emre [1✉], Luis E. Arroyo-García[1], Khanh V. Do[2,3,4], Bokkyoo Jun[2], Makiko Ohshima[1],
Silvia Gómez Alcalde[1], Megan L. Cothern[2], Silvia Maioli[1], Per Nilsson [1], Erik Hjorth[1], André Fisahn [1],
Nicolas G. Bazan [2] & Marianne Schultzberg [1✉]

Sustained microglial activation and increased pro-inflammatory signalling cause chronic inflammation and neuronal damage in Alzheimer's disease (AD). Resolution of inflammation follows neutralization of pathogens and is a response to limit damage and promote healing, mediated by pro-resolving lipid mediators (LMs). Since resolution is impaired in AD brains, we decided to test if intranasal administration of pro-resolving LMs in the $App^{NL-G-F/NL-G-F}$ mouse model for AD could resolve inflammation and ameliorate pathology in the brain. A mixture of the pro-resolving LMs resolvin (Rv) E1, RvD1, RvD2, maresin 1 (MaR1) and neuroprotectin D1 (NPD1) was administered to stimulate their respective receptors. We examined amyloid load, cognition, neuronal network oscillations, glial activation and inflammatory factors. The treatment ameliorated memory deficits accompanied by a restoration of gamma oscillation deficits, together with a dramatic decrease in microglial activation. These findings open potential avenues for therapeutic exploration of pro-resolving LMs in AD, using a non-invasive route.

[1] Department of Neurobiology, Care Sciences and Society, Division of Neurogeriatrics, Center for Alzheimer Research, Karolinska Institutet, Stockholm, Sweden. [2] Neuroscience Center of Excellence, School of Medicine, Louisiana State University Health New Orleans, New Orleans, LA, USA. [3] Faculty of Medicine, PHENIKAA University, Hanoi 12116, Vietnam. [4] PHENIKAA Research and Technology Institute (PRATI), A&A Green Phoenix Group JSC, No.167 Hoang Ngan, Trung Hoa, Cau Giay, Hanoi 11313, Vietnam. ✉email: crnemre@gmail.com; Marianne.Schultzberg@ki.se

Alzheimer's disease (AD), the most common form of dementia, is a progressive neurodegenerative disorder that still lacks an effective treatment. Neuropathological AD hallmarks are deposition of β-amyloid (Aβ) peptide and hyper-phosphorylation of tau, resulting in plaque formation and neu-rofibrillary tangles, respectively[1]. AD pathogenesis also displays neuroinflammation as evidenced by microglial activation and increased levels of proinflammatory mediators[2,3]. Persistent activation of glial cells by excessive Aβ peptide and tissue damage causes continuous production of proinflammatory cytokines, chemokines and lipid mediators (LMs), which recruit and further activate immune cells, fostering a vicious circle[4]. Already at its initiation, the inflammatory response induces the biosynthesis of a distinct group of LMs that promote resolution of the inflam-mation and return to homeostasis by stimulating clearance of cellular and molecular debris, and tissue repair. These pro-resolving LMs[5] include resolvins (Rv), protectins (PD), and maresins (MaR), derived from the omega-3 polyunsaturated fatty acids (PUFAs), docosahexaenoic acid (DHA), and eicosapentae-noic acid (EPA)[6,7]. In addition, lipoxin A$_4$ (LXA$_4$), a pro-resolving LM, is derived from the omega-6 fatty acid arachidonic acid (AA)[8]. These LMs exert their beneficial effects upon binding to membrane-bound G-protein coupled receptors (GPCRs), shown to be expressed in the brain, such as chemerin-like receptor 1 (ChemR23) and leukotriene B4 receptor (BLT1)[9], both of which respond to RvE1, activate protein kinases and block nuclear factor (NF)-κB activation[10]. RvD1 interacts with N-formyl peptide receptor 2 (FPR2) to signal cell survival and phagocytosis[11], and also binds to G-protein coupled receptor 32 (GPR32) reducing TNF-α-stimulated NF-κB response[11]. RvD2 binds to GPR18 to reduce IL-1β secretion and inflammasome activation, shifting immune cells to a pro-resolving phenotype[12]. MaR1 activates leucine-rich repeat-containing G-protein coupled receptor (LGR6)[13] and enhances macrophage phagocytosis of dying cells that is crucial for the removal of tissue debris during resolution[7]. Neuroprotectin D1 (NPD1) binds to G-protein coupled receptor 37 (GPR37), inducing intracellular Ca$^{2+}$ increase and triggering phagocytosis via AKT (protein kinase B) and extracellular signal-regulated kinase (ERK) signalling pathways[14].

The chronic inflammation in the AD brain implies a dys-functional resolution. Indirect evidence for this comes from analyses of *post mortem* brain and blood samples from AD patients showing reduced levels of the precursors of pro-resolving LMs, PUFAs[15]. Direct evidence for a dysfunctional resolution of inflammation in AD is found in decreased levels of pro-resolving LMs in *post mortem* brains from AD patients, compared to healthy individuals[16–18]. However, while pro-resolving LMs are decreased, evidence from our recent studies demonstrated that the levels of receptors shown to interact with LMs are increased in AD brains[17,19]. Attempts to stimulate resolution as a therapeutic target in human AD are few. One clinical trial using PUFAs for the treatment of AD patients showed that omega-3 fatty acids (FAs) were only helpful in patients with mild cognitive impairment[20]. As precursors for several LMs, PUFAs can be catalysed into either pro-inflammatory or pro-resolving LMs, depending on the activity of biosynthetic enzymes, signalling kinases and intracellular Ca$^{2+}$ levels. Hypothetically, the pro-duction of SPMs from PUFAs can be hindered by detrimental influence on their pathway of synthesis exerted by the pathology in the AD brain. Therefore, direct treatment with pro-resolving LMs could more effectively trigger the beneficial cellular and molecular events of resolution. Treatment with pro-resolving LMs in experimental models have shown positive results. In in vitro AD cellular models, neuroprotection and stimulation of Aβ$_{42}$ phagocytosis have been demonstrated[16,18,21]. In vivo studies on transgenic AD mouse models and a model based on intra-hippocampal injection of Aβ have shown beneficial effects of pro-resolving LMs[22–24]. Each pro-resolving LM has a specific profile of biological functions, inducing different protective cellular mechanisms aimed at terminating and resolving the inflammation and a combination therapy may be more effective than a monotherapy. Hence, in the present study, we aimed to investigate whether treatment with five different pro-resolving LMs (RvE1, RvD1, RvD2, MaR1 and NPD1) using intranasal administration in the $App^{NL-G-F/NL-G-F}$ knock-in mouse model of AD positively modulates the memory deficit and neuroinflammation.

## Results
The effects of intranasal administration of five different pro-resolving LMs (RvE1, RvD1, RvD2, MaR1 and NPD1) on memory deficit and neuroinflammation in the homozygotic $App^{NL-G-F}$ mouse model were investigated as described in Methods.

**Pro-resolving LMs rescued memory deficits in $App^{NL-G-F}$ mice.** To examine the effects of intranasal treatment with pro-resolving LMs on cognition and memory in $App^{NL-G-F}$ mice, several beha-vioural tests were employed. Recovery of learning and memory was analysed by the novel object recognition (NOR) and fear conditioning (FC) tests. In the recognition session of the NOR test using two different objects (one novel and one familiar), the $App^{NL-G-F}$-Veh mice failed to discriminate between the familiar and novel object, as shown by a novel object discrimination index (DI) lower than for the WT-Veh group (Fig. 1b), although the difference was not statistically significant. Compared to the $App^{NL-G-F}$-Veh mice, the mice treated with LMs ($App^{NL-G-F}$-LMs) explored the novel object for a longer time period and made a higher number of contacts with the object, as indicated by a higher DI showing that they recalled the familiar object (Fig. 1b). The LM-treated $App^{NL-G-F}$ mice also spent more time exploring the novel object than the familiar object (Fig. 1b), in contrast to $App^{NL-G-F}$-Veh mice that spent an equal time with the familiar and novel object (Fig. 1b).

Contextual and cued FC were performed as described in Fig. 1c (and in Methods). On day 2, all three groups of mice showed increased freezing time compared to day 1, demonstrating that all groups were able to recognize the context that was associated with the foot shock (Fig. 1d). There was no difference in time between groups on day 2. On day 3, however, the $App^{NL-G-F}$-Veh group exhibited reduced freezing time compared to WT-Veh mice, showing that their recollection of the adverse stimulus was impaired. In $App^{NL-G-F}$ mice treated with LMs ($App^{NL-G-F}$-LMs) the ability to recollect was spared, as shown by a memory function reaching the capacity of the WT mice, indicated by longer freezing time compared to the $App^{NL-G-F}$-Veh mice (Fig. 1d).

Exploratory activity, general motor functions and anxiety-like behaviour were investigated by testing the mice in open field (OF) and elevated plus maze (EPM). In the OF test, there was no statistically significant difference between the groups with regard to the total distance travelled (Fig. 1e). However, the locomotor activity was lower in $App^{NL-G-F}$-Veh compared to WT-Veh mice, while there was no difference in activity between the WT-Veh mice and the $App^{NL-G-F}$ mice treated with LMs (Fig. 1e). Thigmotaxis was investigated by assessing the time spent in the outer and centre zone of the arena. The treatment with LMs did not significantly affect the locomotor activity or thigmotaxis as compared to vehicle-treated WT or $App^{NL-G-F}$ mice.

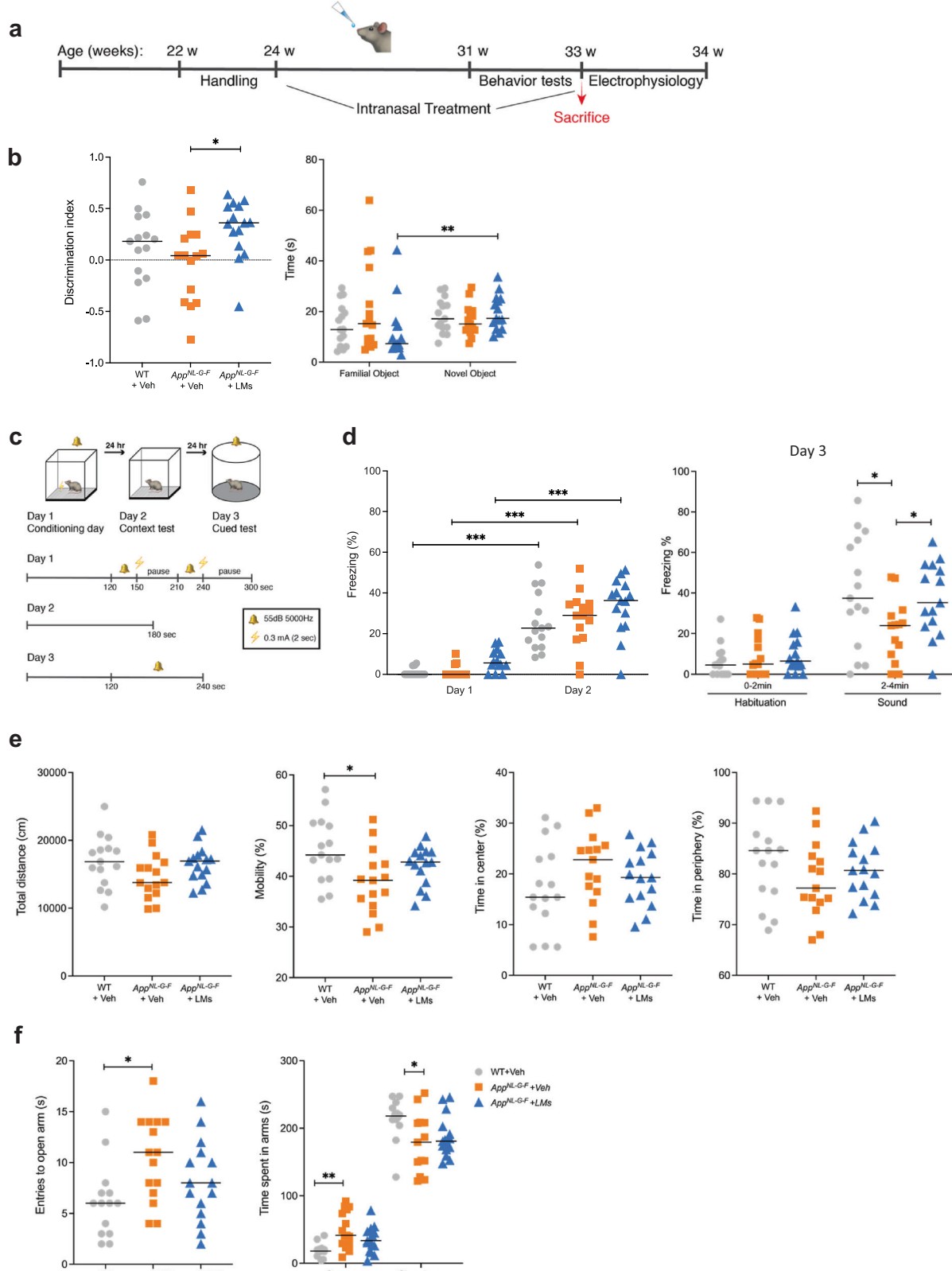

In the EPM test, the $App^{NL-G-F}$-Veh mice spent longer time in the open arms and made an increased number of entries into the open arms compared to WT mice, indicating reduced anxiety-like behaviour (Fig. 1f). The $App^{NL-G-F}$ mice that were treated with LMs spent the same time in both the open and closed arm as the WT mice. Similar results were obtained by analysis of the time passing before the first visit to the closed arms (Fig. 1f).

**Pro-resolving LMs rescued hippocampal gamma oscillations in $App^{NL-G-F}$ mice**. The $App^{NL-G-F}$ mouse model exhibits impairment of hippocampal gamma oscillations at a very early stage of the pathology[25]. The present data confirmed deterioration of in vitro gamma oscillations at 8 months of age in the $App^{NL-G-F}$ mouse model compared to WT mice (Fig. 2a–c). Importantly, the treatment with LMs showed recovery of about 57% of gamma

**Fig. 1 Effects of LMs on learning and memory tests, locomotor activity and anxiety-like behaviour in control and $App^{NL-G-F}$ mice. a** Experimental design of treatment with pro-resolving lipid mediators (LMs) in $App^{NL-G-F}$ mice ($App^{NL-G-F}$-LMs) and administration of vehicle (0.9% saline) in $App^{NL-G-F}$ ($App^{NL-G-F}$-Veh) and wild-type (WT-Veh) mice. **b** Novel object recognition (NOR) test showed that treatment with LMs resulted in increased discrimination index (DI) in the $App^{NL-G-F}$ mice ($App^{NL-G-F}$-LMs). The score zero and higher indicates more time spent with the novel object than with the familiar object. The LM-treated group ($App^{NL-G-F}$-LM) displayed an increased exploration time for the novel object, whereas the other groups (WT-Veh and $App^{NL-G-F}$-Veh) did not show such a difference. **c** Experimental design and fear training scheme for the fear conditioning (FC) test. **d** All three animal groups showed increased percentage of freezing time between the habituation phase (day 1) and the context testing (day 2), but there was no difference between the groups, nor in the freezing in the cued test (day 3). However, the $App^{NL-G-F}$-Veh group displayed a lower percentage of freezing time upon exposure to sound during day 3 compared to the WT-Veh group, and the treatment with LMs restored this to control levels in the $App^{NL-G-F}$ mice ($App^{NL-G-F}$-LMs). **e** Locomotor activity and explorative behaviour were analysed in the open field (OF) test. There was no difference between the treatment groups regarding total distance covered, mobility or time spent in the centre. **f** Elevated plus maze (EPM) test for anxiety-like behaviour showed an increased number of entries into and time spent in the open arms, and a decrease in the time spent in the closed arms by the $App^{NL-G-F}$ mice given vehicle ($App^{NL-G-F}$-Veh) compared to WT mice (WT-Veh), whereas no effect was observed by treatment with LMs ($App^{NL-G-F}$-LMs). Comparisons between treatment groups were performed with Kruskal-Wallis with Dunn's multiple comparisons *post hoc* test, *$P < 0.05$, **$P < 0.01$, ***$P < 0.001$, ****$P < 0.0001$ ($n = 14$-$15$ mice/group). Results are presented as median in scatter plots.

oscillation power in the $App^{NL-G-F}$ mouse model (Fig. 2a, b). In contrast, the treatment had no effect on gamma oscillation frequency variance (Fig. 2a–c).

$App^{NL-G-F}$ mice exhibit firing desynchronization of action potential in fast-spiking interneurons (FSN) long before desynchronization of pyramidal cells and plaque formation[25]. Our results showed that the pathology in the $App^{NL-G-F}$ mouse model affects FSNs (Fig. 2d–i). The $App^{NL-G-F}$-Veh mice showed a decrease in the excitatory postsynaptic current (EPSC) charge transfer in FSNs compared to WT mice (Fig. 2d, e). Notably, the treatment with LMs resulted in the recovery of the EPSC charge transfer (Fig. 2d, e).

The membrane potential (Em) of FSNs was analysed upon network activation with kainic acid (KA) (100 nM) as used to induce gamma oscillations. However, there was no statistically significant difference between the groups (Fig. 2f).

Comparison of the synchronization across the different groups showed that the LM treatment in the $App^{NL-G-F}$ mice partially recovered the desynchronization found in the $App^{NL-G-F}$-Veh group (Fig. 2g, h). Moreover, we found that this partial recovery in the synchronization was independent of the FSN firing rate, which did not change significantly (Fig. 2i).

**Pro-resolving LMs attenuated microgliosis in $App^{NL-G-F}$ mice.** The activation of microglia and astrocytes was examined by immunohistochemistry for Iba1 (Fig. 3a, b) and GFAP (Supplementary Fig. 1a, b). The $App^{NL-G-F}$-Veh mice displayed abundant Iba1-positive cells that had an enlarged ameboid morphology and were clustered together, whereas the WT mice had ramified microglia with smaller cell bodies that were more evenly distributed in the tissue (Fig. 3a). In $App^{NL-G-F}$ mice treated with LMs the ameboid morphology and clustering of microglia appeared to be reduced (Fig. 3a). Quantitative assessment by densitometry (Fig. 3b) confirmed these results by showing a pronounced increase in microglial activation in both cortex and hippocampus of $App^{NL-G-F}$-Veh mice as shown by a significantly larger area of Iba1 immunoreactivity compared to WT mice, and a marked reduction in the $App^{NL-G-F}$ mice treated with LMs (Fig. 3b).

Analysis of astrocyte activation showed a marked increase in GFAP staining in the $App^{NL-G-F}$ mice compared to the WT mice, but there was no significant reduction seen upon treatment with LMs (Supplementary Fig. 1a, b).

**Amyloid burden not affected by pro-resolving LMs.** To examine whether the pro-resolving LM treatment altered the amyloid plaque load in the $App^{NL-G-F}$ mice, the number of diffuse and neuritic plaques that were labelled with both Thioflavin-S and antibodies for Aβ peptide was counted in the cerebral cortex and hippocampus (Fig. 3d). Neither diffuse nor neuritic plaque counts were altered by the LM treatment (Fig. 3e). We also analysed the levels of Aβ$_{42}$ by immunoassay in formic acid extracts of brain homogenates (Fig. 3f). There was no effect of treatment with LMs on the Aβ$_{42}$ levels compared to vehicle (Fig. 3f), consistent with the immunohistochemical analysis.

**Effects of pro-resolving LMs on expression markers for resolution of inflammation and of neurotransmission.** To investigate factors contributing to the recovery in memory function we used Western blot to analyse proteins previously shown to mediate effects of LMs, as well as synaptic markers and a marker for phagocytic microglia (Fig. 4a, Supplementary Fig. 2). We analysed proteins shown to mediate the activities of pro-resolving LMs in the periphery. Thus, the levels of BLT1 and ChemR23 in cerebral cortex were lower in $App^{NL-G-F}$ mice compared to the basal levels in the WT-Veh group. However, treatment with LMs did not affect the levels of these two receptors nor of LGR6, GPR18 or FPR2 in the $App^{NL-G-F}$-LM group in comparison to the $App^{NL-G-F}$-Veh group (Fig. 4b). In the hippocampus, there was no difference in these receptor proteins between WT and $App^{NL-G-F}$ mice, nor upon treatment of $App^{NL-G-F}$ mice with LMs (Fig. 4c). However, an increase in LGR6 was observed between WT and $App^{NL-G-F}$ mice treated with LMs (Fig. 4c).

Analysis of receptors for the neurotransmitters gamma-aminobutyric acid (GABA) and glutamate showed no difference between WT and $App^{NL-G-F}$ mice with regard to glutamate receptor 1 (GLUR1) and 4 (GLUR4), and the GABA$_A$ receptor subunit 1α (GABA$_A$1α), neither in cerebral cortex nor hippocampus (Fig. 4b, c). In the hippocampus, the levels of GLUR1 and GLUR4 were lower in $App^{NL-G-F}$ mice treated with LMs than in WT mice, but there was no difference between the $App^{NL-G-F}$-Veh and $App^{NL-G-F}$-LM group (Fig. 4c). The levels of GABA$_A$1α were decreased in $App^{NL-G-F}$-LM compared to the $App^{NL-G-F}$-Veh mice in both cortex and hippocampus (Fig. 4b, c).

The $App^{NL-G-F}$-Veh mice had lower cortical levels of the postsynaptic marker PSD95 compared to WT-Veh mice (Fig. 4b), but treatment with LMs did not affect this difference. There was no difference in PSD95 levels in the hippocampus between the groups (Fig. 4c).

Levels of the microglial activation marker TREM2 were higher in both cortex and hippocampus of $App^{NL-G-F}$-Veh mice compared to the WT control (WT-Veh) group, whereas effects upon treatment with LMs did not reach statistical significance (Fig. 4b, c).

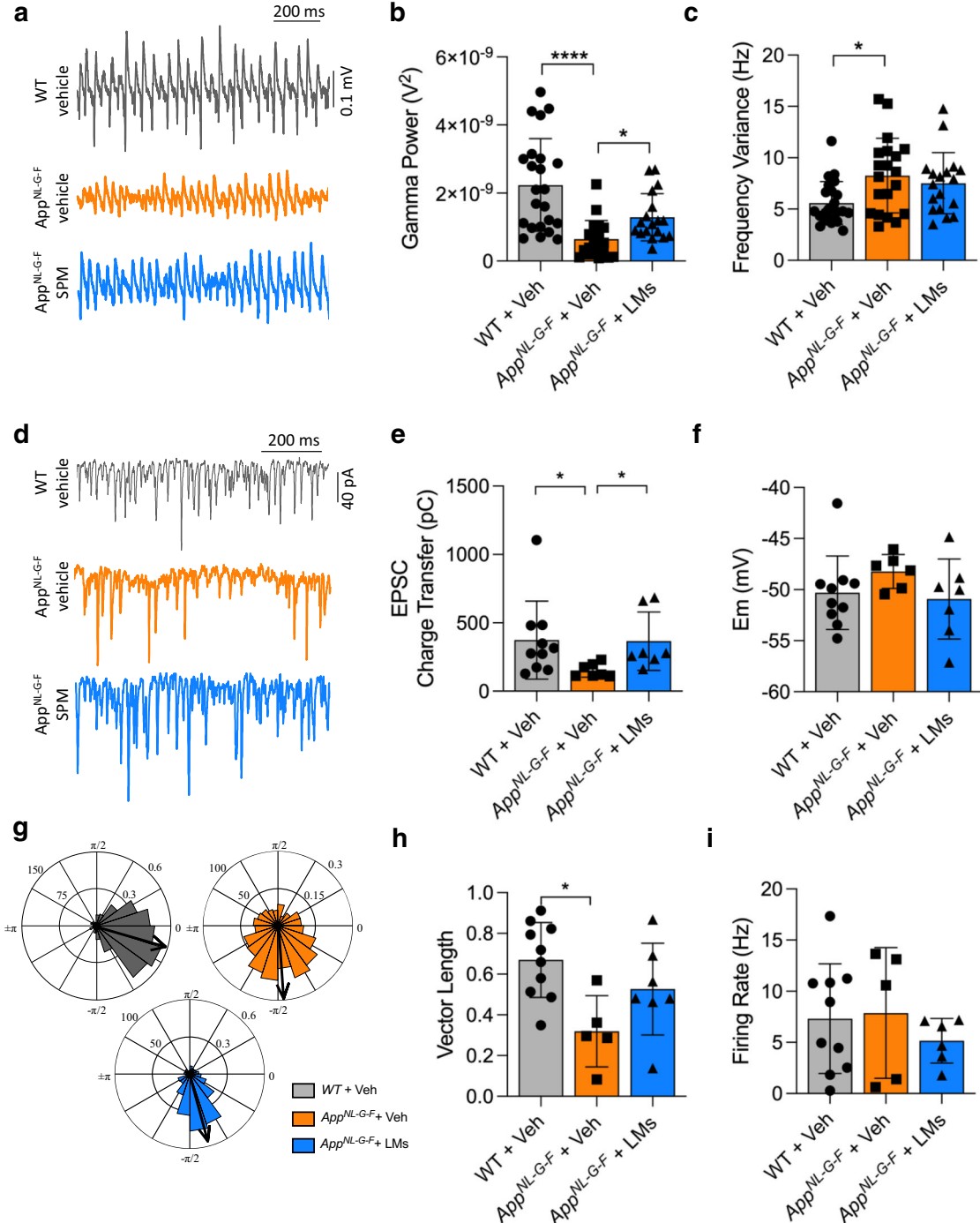

**Fig. 2 LMs partially restore gamma oscillation power and synchronization in *App^NL-G-F* mice. a** Representative traces of gamma oscillations in hippocampal brain slices are shown in wild-type (WT) and *App^NL-G-F* mice given vehicle (Veh) and *App^NL-G-F* mice treated with lipid mediators (LMs). **b** Bar graphs of the gamma oscillation power in the three animal groups. Treatment with LMs partially restored the reduced gamma power observed in *App^NL-G-F* mice. **c** Bar graphs of gamma oscillations described in **b**. There was an increase in frequency variance in the *App^NL-G-F* mice, but the LM treatment did not reduce this. **d** Representative traces of excitatory postsynaptic currents (EPSCs) from fast-spiking interneurons (FSNs) in each condition. **e** Bar graphs showing EPSC charge transfer in FSNs from each condition. Treatment with LMs restored the reduction in EPSC charge transfer observed in *App^NL-G-F* mice. **f** Bar graphs of FSN membrane potential (Em) from each condition. There was no difference in the Em between the three groups. **g** Polar-plots showing the distribution of AP phase-angles from the experimental conditions described in A. **h** Bar graphs of resulting vector length derived from concomitant recordings of gamma oscillations and FSN (AP) distribution for each experimental condition. There was a decrease in the vector length in the *App^NL-G-F* mice, but the LM treatment did not restore this. **i** Bar graphs of FSN firing rate obtained from H. Kruskal-Wallis and one-way ANOVA tests were used for group comparisons. Data are presented as mean ± S.E.M. *$P < 0.05$, **$P < 0.01$, ***$P < 0.001$.

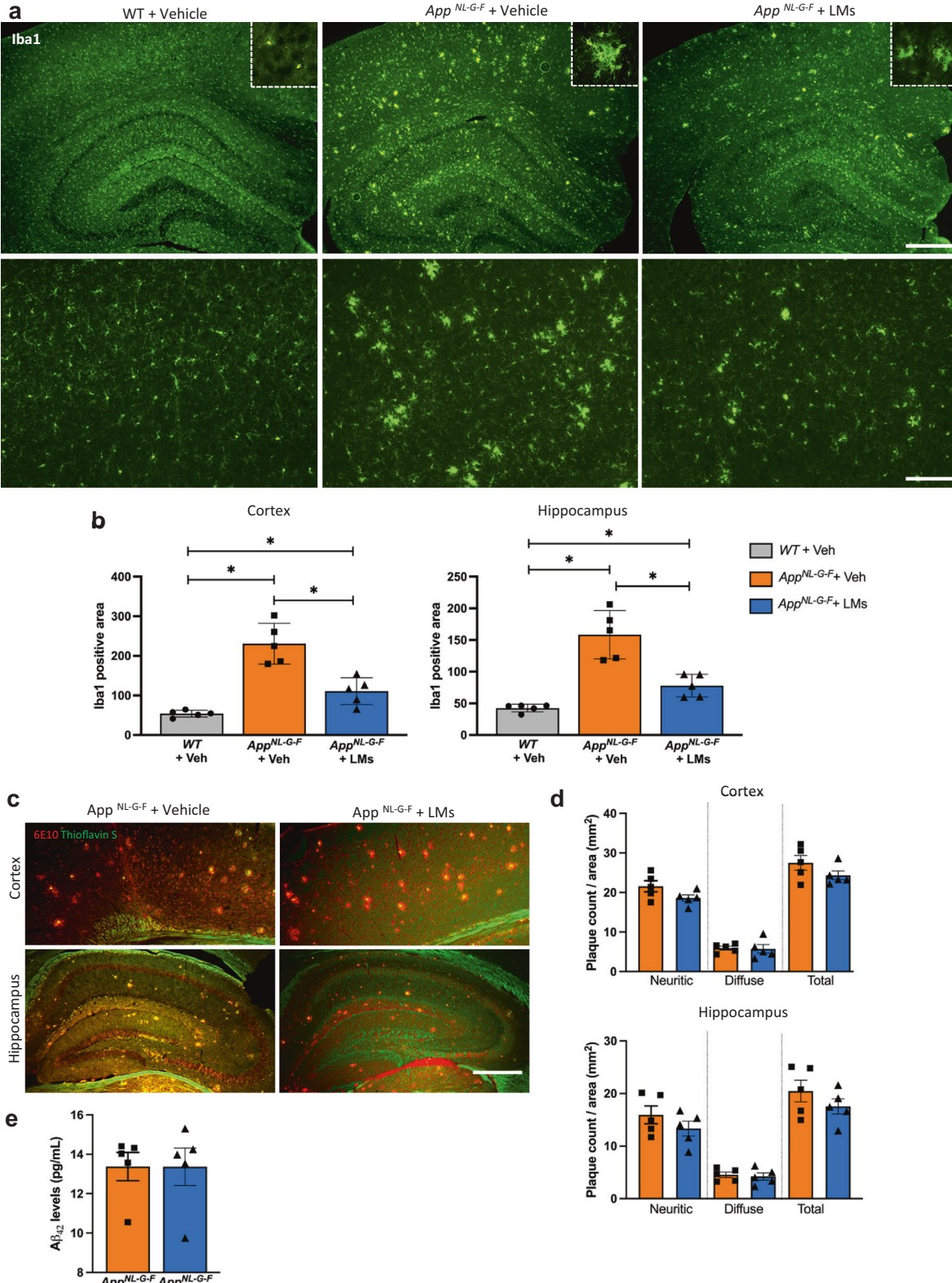

**Cytokine and chemokine levels remain unchanged in the cerebral cortex and hippocampus after LM treatment**. Next, we investigated the effects of LMs on pro- and anti-inflammatory cytokines and chemokines using multi-immunoassays (Fig. 5 and Supplementary Fig. 3). In the cortex, the levels of interferon-γ-induced protein 10 (IP-10), macrophage inflammatory protein (MCP)-1, MIP-1α and MIP-2 were higher in

$App^{NL-G-F}$-Veh mice compared to WT-Veh mice (Fig. 5). In the hippocampus, the levels of interleukin (IL)-1β, keratinocyte chemoattractant/human growth-regulated oncogene (KC-GRO), IL-15, IL-33, IP-10, MCP-1, MIP-1α and MIP-2 were higher in $App^{NL-G-F}$-Veh mice than in WT-Veh mice (Fig. 5). Although the data hinted at a decrease in the levels of most of these factors upon treatment of $App^{NL-G-F}$ mice with LMs, none

**Fig. 3 Effects of LMs on microglial activation and amyloid burden. a** Immunohistochemical staining of brain sections of cortical and hippocampal areas from $App^{NL-G-F}$ and wild-type (WT) mice using the microglia marker Iba1, showing higher numbers of Iba1-labelled microglia in $App^{NL-G-F}$ mice than in both WT mice and $App^{NL-G-F}$ mice treated with LMs. **b** Densitometry of the area covered by Iba1-positive microglia in WT-Veh, $App^{NL-G-F}$-Veh and $App^{NL-G-F}$-LMs mice showing the differences seen in **a. c** Immunohistochemistry for APP and Thioflavin-S co-staining in the cerebral cortex and hippocampus from WT-Veh, $App^{NL-G-F}$-Veh and $App^{NL-G-F}$-LMs mice. **d** Comparison of the counts of diffuse and neuritic Aβ plaques in the cortex and hippocampus shows no difference between the $App^{NL-G-F}$-Veh and $App^{NL-G-F}$-LMs group. **e** Analysis of Aβ$_{42}$ levels in formic acid extracts of cerebral cortex by Meso Scale did not show any effect of LM treatment in $App^{NL-G-F}$ mice. Mann-Whitney U test was used for comparisons and correction was performed manually by multiplying with the number of comparisons. Data are presented as mean ± S.E.M ($n = 5$ mice/group). *$P < 0.05$, **$P < 0.01$, ***$P < 0.001$. Scale bars = 600 and 150 μm. Iba1 ionized calcium binding adaptor protein 1.

of these differences were statistically significant (Fig. 5 and Supplementary Fig. 3).

**DHA-containing phospholipids increase after LM treatment.** Phospholipid content and composition were analysed by liquid chromatography tandem-mass spectrometry (LC-MS/MS) to investigate the effects of LM treatment in the $App^{NL-G-F}$ mice. There was no difference between $App^{NL-G-F}$ mice and WT controls. However, there was an increase in the DHA-containing phosphatidylserine (PS) species 16:0/22:6 and 22:6/22:6 in the hippocampus of $App^{NL-G-F}$ mice treated with LMs compared to the $App^{NL-G-F}$-Veh group (Fig. 6). Moreover, phosphatidylcholine (PC) 42:9, phosphatidyletanolamine (PE) 46:8, PS 44:10, PS 40:4, and PS 44:1 were increased in the hippocampus and PS 44:4 was higher in the cortex of $App^{NL-G-F}$-LM mice compared to $App^{NL-G-F}$-Veh mice (Fig. 6).

The levels of free PUFAs, intermediate metabolites, and pro-inflammatory and pro-resolving LMs were also analysed (Supplementary Fig. 4), however, there were no differences between the groups investigated in this study.

The deuterium-labelled LMs administered in two of the $App^{NL-G-F}$ mice were detectable by LC-MS/MS (Supplementary Fig. 5) and support that the intranasally administered LMs reach the brain where they can exert their functions.

**Discussion**
AD pathology is characterized by the initial accumulation of oligomeric Aβ peptides that aggregate, and of neurofibrillary tangles (NFTs), leading to oxidative stress, neurodegeneration, synaptic loss, and cognitive impairment. Chronic inflammation has for long been regarded as a consequence of the major pathological hallmarks of AD[26], but several studies from recent years suggest that inflammation may precede cognitive symptoms, as well as plaque and tangle formation. The inherent pro-inflammatory nature of Aβ together with its neurotoxicity causes long-lasting activation of microglia, astrocyte proliferation, and sustained release of inflammatory factors, contributing to further Aβ production[4]. Despite epidemiological studies indicating that anti-inflammatory drugs reduce the prevalence of AD, these treatment strategies have failed, and even resulted in side effects[27]. Therefore, the approach of blocking AD inflammation does not seem viable.

Neuroinflammation is a complex and dynamic process during which glial cell activities are both protective and detrimental. Resolution is the end stage of an inflammatory response, highly coordinated and mediated by pro-resolving LMs[6], and involves many protective activities meant to downregulate inflammation and restore homeostasis in the tissue. Impairment of the resolution of inflammation in the human AD brain is supported by reduced levels of proresolving LMs[16–18] and alterations of their receptors[17,19]. Furthermore, proresolving LMs have been shown to stimulate Aβ phagocytosis and reduce the pro-inflammatory phenotype in in vitro AD models[18,21], and to shift Aβ processing

from an amyloidogenic to a non-amyloidogenic pathway[24]. Combined administration of LXA$_4$ and RvE1 reduced plaque load and cytokine levels in the transgenic 5xFAD mouse model[22] and treatment with RvE1 in a Down syndrome model prevented memory loss and neuroinflammation[28]. In an AD mouse model based on intrahippocampal injections of Aβ the intracerebroventricular (icv) administration of MaR1 reduced cognitive decline and attenuated microglia and astrocyte activation[23]. Studies on epilepsy and traumatic brain injury models demonstrated a beneficial effect on memory and learning upon treatment with NPD1 or resolvins[29,30]. Incubation of microglial cells with RvD1 and RvE1 upon LPS treatment promoted resolution, showing decreased pro-inflammatory cytokine levels and NF-κB activation[31]. Therefore, promoting the progression of inflammation to its pro-homeostatic end stage, resolution, may be a more viable approach than merely stopping it.

LMs are fragile molecules and prone to oxidation which makes peroral administration unlikely to be effective. On the other hand, invasive injection-based administration is not desirable for a treatment that may be effective if given in an outpatient daily or semi-daily basis. Here, the non-invasive intranasal route of administration of a combination of five different LMs clearly resulted in beneficial effects in the $App^{NL-G-F}$ mouse model in terms of behaviour, cognition, neuroinflammation as well as neuronal network aspects of memory. The $App^{NL-G-F}$ mouse model expresses APP at wild-type levels, producing a robust increase in Aβ levels at 2 months of age and showing cognitive impairment starting at 6-months-of-age[32]. Our results show that treatment with LMs improved recognition memory in $App^{NL-G-F}$ mice and produced a significant restoration of the deficits this AD model exhibits regarding contextual fear memory reconsolidation and the formation of cued fear memories. The effects of LMs on cognition appear uncoupled from potential confounding effects on locomotor functions and coordination, or from stress and anxiety as shown by the lack of effects in the OF and EPM tests, respectively. Different parameters were analysed in order to provide insight into possible mechanisms of the effects observed and the data are discussed in the following.

The $App^{NL-G-F}$ mouse model exhibits progressive deterioration of cognition-relevant electrical brain rhythms, especially in the gamma frequency band (30-80 Hz)[25,33]. Gamma oscillations are evoked during cognitive processes and are the result of rhythmic synchronization of excitatory and inhibitory currents in the relevant neuronal networks. In the hippocampus, gamma oscillations play a role in memory formation and recall, as well as in learning behaviours[34]. Recent evidence suggests that the oligomeric Aβ peptide (non-aggregated) interferes with proper network synchronization in the hippocampus, leading to learning and memory impairment[35]. The power and synchrony of the gamma brain rhythm depend on the proper firing synchronization of GABAergic FSNs and glutamatergic pyramidal cells[34]. The GABAergic FSNs play a critical role in the recovery of gamma oscillations after impairment[36]. The present study showed a rescuing effect of LMs on the in vitro gamma oscillations in the

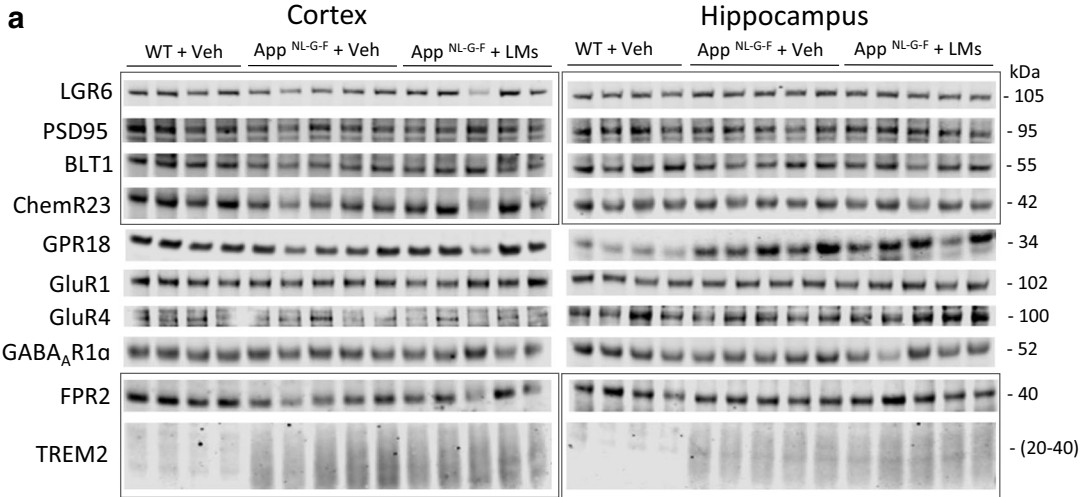

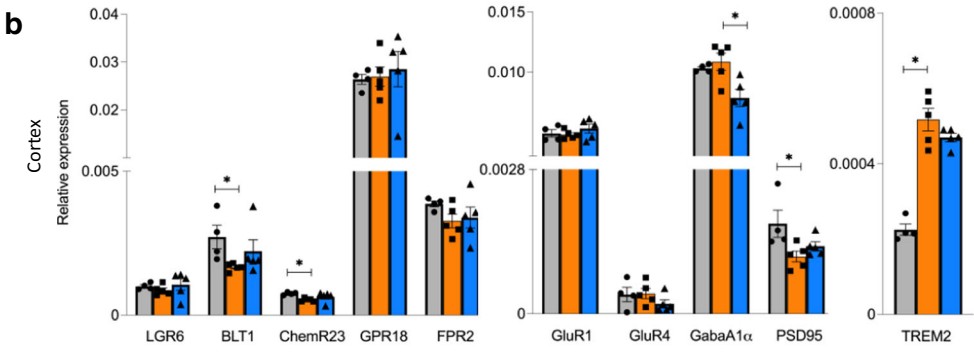

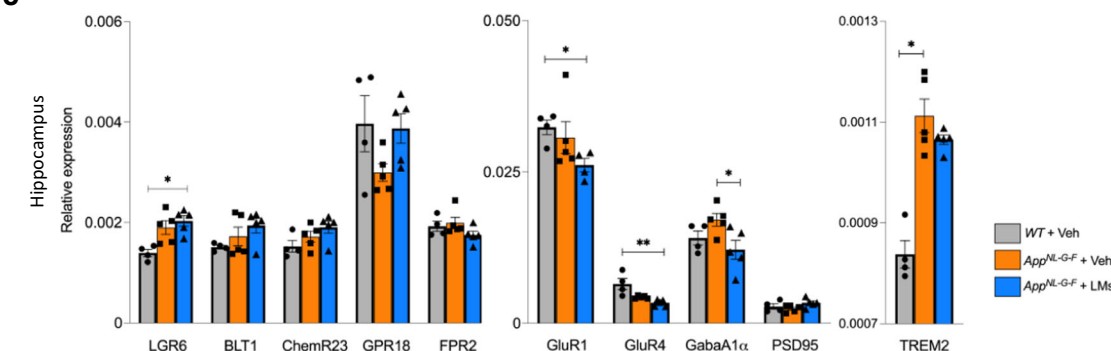

**Fig. 4 Receptors and synaptic markers are altered in $App^{NL-G-F}$ mice. (a–c)** Analysis of receptors for pro-resolving lipid mediators (LMs) (LGR6, BLT1, ChemR23, GPR18 and FPR2), glutamate and GABA receptors (GluR1, GluR4, GABA$_A$1α), a synaptic marker (PSD95), and an inflammation marker (TREM2), was performed by Western blot in cortex and hippocampus of WT + Veh, $App^{NL-G-F}$ + Veh and $App^{NL-G-F}$ + LMs mice. Representative blots (**a**) and quantitative analysis (**b, c**) are shown (markers from the same blots are indicted by rectangles, and complete Western blots are shown in Supplementary Fig. 2). Data are presented as mean ± SEM and analysed by Kruskal-Wallis one-way analysis of variance test with Dunn's *post hoc* test (*$P < 0.05$, **$P < 0.01$, ***$P < 0.001$, ****$P < 0.001$) between treatment groups for each marker ($n = 4 - 5$ mice/group). Data are presented as mean ± S.E.M. LGR6 leucine-rich repeat containing G-protein coupled receptor 6, BLT1 leukotriene B4 receptor, ChemR23 chemokine-like receptor 1, GPR18 G-protein-coupled receptor 18, FPR2 formyl peptide receptor 2, TREM2 triggering receptor expressed on myeloid cells 2, GluR1 glutamate receptor 1, GluR4 glutamate receptor 4, GABA$_A$1α gamma-aminobutyric acid A receptor subunit 1α, PSD95 postsynaptic density protein 95, FC fear conditioning, NOR novel object recognition, DI discrimination index.

$App^{NL-G-F}$ mice, indicating that the LM treatment was able to restore the neuronal conditions for excitatory input to FSNs.

The balance between excitation and inhibition in neuronal networks contributes to the synchrony of action potential firing. The majority of excitatory and inhibitory signals are mediated *via* glutamate and GABA receptors, respectively. There are several factors influencing the regulation of these synaptic inputs since the expression pattern for these receptors and their functions are brain region-specific[37]. GABA$_A$ receptors are ionotropic receptors containing five subunits surrounding a Cl$^-$ channel that is activated by GABA, which further induces Cl$^-$ influx and inhibits excitatory stimulation[38]. In mouse brain, the GABA$_A$ receptor

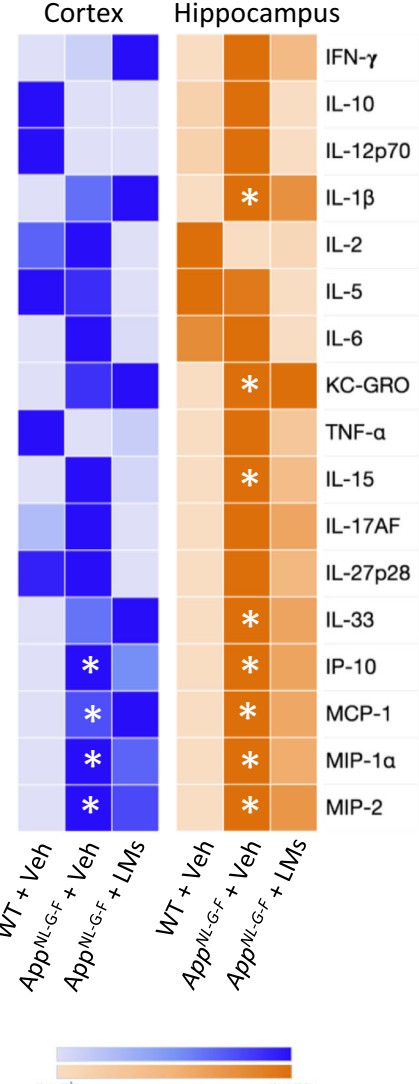

**Fig. 5 Cytokines and chemokines are altered in the brain of *App^NL-G-F* mice.** Cytokines and chemokines were analysed in homogenates of cerebral cortex and hippocampus by multi-immunoassay (Meso Scale v-plex). Rows in the heatmap represent cytokines and chemokines and columns represent treatment groups. The colours represent median of normalized concentration values. $n = 5$ mice/group. Asterisks denote statistical significance between wild-type (WT) and *App^NL-G-F* mice given vehicle (Veh) using Kruskal-Wallis with Dunn's *post hoc* test. IFN-γ interferon-γ, IL interleukin, IP-10 interferon-γ-induced protein 10, KC-GRO keratinocyte chemoattractant/human growth-regulated oncogene, MCP-1 monocyte chemoattractant protein, MIP macrophage inflammatory protein, TNF-α tumour necrosis factor-α.

septum. Indeed, twenty GABA$_A$ receptor have been identified and their pentameric composition determines cellular localization and function[37], demonstrating the complexity of this system. Recently, alterations in GABA$_A$ receptor subunits were shown to be specific for the brain region and cell layer in the human AD brain[42]. Therefore, changes in these receptors are complex and difficult to interpret, but the changes we observed may indicate the importance of a stable neuronal network and reorganization of neuronal circuits in the presence of AD pathology.

GLUR1 and GLUR4 are subunits of α-amino-3-hydroxy-5-methyl-4-isoxazole (AMPA) receptors, which are ionotropic glutamate receptors coupled to ion channels to regulate excitatory signals[43]. Similar to GABA, these AMPA receptor subunits have a region-specific distribution within the brain, indicating that their functions rely on the subunit complex and on the localization[44]. Oligomeric Aβ peptide was shown to disrupt AMPA receptor trafficking by inhibition of GLUR1 and GLUR4 delivery to the synaptic endings in an AD model[45]. RvE1 and RvD1 have been demonstrated to normalize the synaptic plasticity by reducing TNF-α-induced N-methyl-D-aspartic acid (NMDA) receptor hyperactivity by inhibition of the ERK signalling pathway[46]. Decreased activity of NMDA receptors results in decreased AMPA receptor trafficking[47], however, there was no effect by LM treatment on the levels of GLUR1 or GLUR4 in the *App^NL-G-F* mice, which does not exclude changes in receptor affinity.

The post-synaptic marker PSD95 is a membrane-associated guanylate kinase that regulates the surface expression of glutamate receptors[48]. PSD95 levels are decreased in the human AD brain, in correlation with disease pathology[49]. A decrease in PSD95 levels was observed in the cortex of *App^NL-G-F* mice compared to WT mice, but treatment with pro-resolving LMs did not affect the PSD95 levels in cortex nor hippocampus. Evidence for beneficial effects of pro-resolving LMs on neuronal connectivity was provided in a study on the APP/PS1 mouse model demonstrating a protective role of pro-resolving LMs by restoring the levels of microtubule-associated protein 2 (MAP2), synapsin 1 and PSD95 upon treatment with N-acetyl sphingosine that acetylates COX-2 and triggers the production of pro-resolving LMs[50]. Altogether, previous studies and the behavioural and electrophysiological data described here support the beneficial effects of LMs on memory in the pathology of AD. Further studies are required to elucidate the mechanisms of action of these effects, and the presence of LM receptors in neurons[17,19] indicate that direct effects on neurons are indeed possible.

Microglia are the first responders to injury and disease pathology in the brain. They are involved in tissue inflammation and clearance of cell debris[51,52]. Despite the beneficial functions of microglia such as Aβ phagocytosis, their excessive or uncontrolled activation and proliferation result in the release of pro-inflammatory proteins and LMs, which contribute to neuronal cell death. As the key mediators of resolution, proresolving LMs have been shown to activate microglia into a phenotype where they produce higher levels of anti-inflammatory mediators and facilitate the removal of Aβ[18,21,53]. The treatment of *App^NL-G-F* mice with LMs demonstrated a significant decrease in microglial activation as determined by analysis of the area covered by Iba1-positive microglia. Although there is no clear-cut consensus on the relationship between microglial phenotype status and their morphological changes, our data suggest that treatment with LMs induces a less activated phenotype, which is associated with an anti-inflammatory state characterized by resolution. These findings agree with previous reports on the effects of pro-resolving LMs[22–24]. Interestingly, there was no effect on the astrocyte activation in the *App^NL-G-F* mice as determined by the area covered by GFAP-positive astrocytes. In contrast, however, recent studies in a model of traumatic brain injury (TBI) showed that

expression does not change with age[39], and GABAergic neurons seem to be spared from AD degeneration[40]. However, recent work demonstrated the contribution of the GABAergic system in circuit dysfunction due to the presence of amyloid, which causes hyperexcitability[41]. Analysis of the GABA$_A$1α showed no difference between WT and *App^NL-G-F* mice, but there was a significant decrease in both hippocampus and cortex in *App^NL-G-F* mice treated with LMs compared to animals given vehicle. This may indicate that LMs balance the excitatory and inhibitory signals since LMs also resulted in behavioural improvements upon treatment of the *App^NL-G-F* mice. However, the GABAergic inhibitory system is regulated by several factors involved in AD pathology, such as cholinergic input from basal forebrain and

**Fig. 6 DHA-containing phospholipids are elevated by treatment with LMs.** Scatter plots for phospholipids analysed by LC-MS/MS in homogenates of hippocampus and cerebral cortex from wild-type (WT) and $App^{NL-G-F}$ mice given vehicle (Veh) and $App^{NL-G-F}$ mice treated with lipid mediators (LMs). A marked increase in DHA-containing phospholipids in the hippocampus can be seen for phosphatidylserine (PS) 40:7 and 44:12 upon treatment with LMs in $App^{NL-G-F}$ mice compared to WT controls (WT-Veh). In addition, PS 44:10 and PS44:1 are increased in the hippocampus, and PS44:4 is increased in cortex upon the LM treatment. Comparisons between treatment groups were performed with Kruskal-Wallis with Dunn's multiple comparisons *post hoc* test, *$P < 0.05$, **$P < 0.01$, ***$P < 0.001$, ****$P < 0.0001$. Data are presented as mean ± S.E.M. PC phosphatidylcholine, PE phosphatidylethanolamine.

treatment with the pro-resolving LMs LXA$_4$ and RvD$_1$ decreased astrocyte activation by binding to ALX/FPR2 receptor[54,55]. Similarly, the administration of MaR1 in the AD model based on icv injection of Aβ resulted in attenuated astrocyte activation[23]. It should be noted that in these studies the LMs were given directly after the injury corresponding to the acute stage of inflammation, or together with an acute injection of Aβ, whereas the $App^{NL-G-F}$ mice display Aβ deposits starting from 2 months and the treatment with LMs in the present study started 4 months later and could therefore be insufficient to alter advanced astrogliosis.

In a previous study, we demonstrated that the $App^{NL-G-F}$ mice show increased levels of pro-inflammatory cytokines at 8 months of age, and the most marked changes regarding several pro-inflammatory factors occurred at 18 months[56]. Although accumulating evidence indicates reduced release of pro-inflammatory cytokines and chemokines by treatment with proresolving LMs in

both in vitro and in vivo models[23,28,57], the treatment with pro-resolving LMs did not significantly alter the levels of the cytokines or chemokines analysed. This was unexpected in view of the effect on microglia, where a reduction in Iba1 labelling was seen. However, although no statistically significant reduction was found, the levels of several pro-inflammatory cytokines seem to be lower in the treated animals, and a larger number of animals may give a significant answer, thus warranting further investigation.

Our studies on human *post mortem* brains showed that BLT1 and ChemR23, receptors for pro-resolving LMs, were increased in AD[17,19], while in the $App^{NL-G-F}$ model we have found that the levels of BLT1, ChemR23 and LGR6 were unchanged at 8 months of age[56], while the levels of FPR2 were reduced, and the levels of GPR18 elevated[56]. In the present study, there was a reduction in the levels of ChemR23 and BLT1 in $App^{NL-G-F}$ mice compared to

WT mice. Treatment with pro-resolving LMs did not affect the levels of the receptors, except for LGR6, which was increased upon LM treatment in the $App^{NL-G-F}$ mice compared to the levels in naïve WT mice. It is crucial to emphasize that these receptors have many other ligands[58], some even pro-inflammatory, and which induce various downstream signalling pathways and trigger their endocytosis[59], and that the receptor pharmacology of pro-resolving LMs is far from elucidated. Spatially, ChemR23 and BLT1 are expressed in both neurons and glia[17,19] indicating their capability to respond to LMs.

Impaired clearance of Aβ plays a significant role in abnormal Aβ accumulation and plaque formation in AD pathogenesis[60]. Pro-resolving LMs have been shown to change the cellular morphology of immune cells to prepare for phagocytosis of cell debris and microbes by binding to their specific receptors[61]. Interactions between LMs used in the present study and their respective receptors (RvD1-FPR2, RvD2-GPR18, NPD1-GPR37, RvE1-BLT1 and RvE1-ChemR23) have been shown to trigger phagocytic actions in macrophages[10,11,62,63]. Our studies on in vitro human microglial models demonstrated that MaR1 stimulated Aβ phagocytosis[18,21], and it appears that non-phlogistic phagocytic cleaning of the tissue is a cardinal response in resolution. Therefore, it may seem surprising that the plaque burden in the $App^{NL-G-F}$ mice was unchanged upon treatment with LMs in the present study, nor were the levels of insoluble Aβ affected. However, in comparison the Aβ peptide in vitro presents a much easier target than the large number of plaques in the brain of these mice. The hypothesis that soluble Aβ oligomers may play a more important role in causing neuronal damage, cell loss, oscillatory network dysfunction and memory impairment than insoluble forms of Aβ is gaining traction[64,65]. Unfortunately, the levels of soluble Aβ levels were below the detection level in the detergent soluble fraction used for analysis, and thereby it was not possible to determine if treatment with LMs affected oligomeric forms of Aβ before fibrillization and plaque formation. Moreover, further analysis is necessary to establish whether the effects of the LM treatment on cognitive functions and microglial activation are due to reducing oligomeric or monomeric Aβ forms. In view of findings from in vitro studies demonstrating that these bioactive LMs have effects on neurons and microglia it is possible that the effects observed in vivo in the mice are mediated by direct actions on these cells, as supported by the expression of LM receptors in these cells[17,19]. Thereby it is interesting to consider that the effects of these LMs are not only due to their functions to mediate the resolution of inflammation, but also in neuroprotective/neuroactive roles, such as described for NPD1[16,24,66].

TREM2 is important for the regulation of microglial inflammatory responses and phagocytosis[67]. Recent studies discovered gene variants of TREM2 as risk factors for developing AD[68], and there is evidence from AD mouse models that TREM2-mediated microglial activation contributes to uptake and degradation of soluble Aβ at early stages, whereas it contributes to the formation of dense-core plaques at later stages[69,70]. In studies on different ages of the $App^{NL-G-F}$ mouse model, we demonstrated elevated levels of TREM2 at 8 and 18 months of age compared to WT mice[56], supported by the current study on 8 months old $App^{NL-G-F}$ mice, showing elevated TREM2 levels in both hippocampus and cortex but no treatment effect by LMs. Further studies are required to evaluate whether TREM2 is involved in mediating the effects of LMs on memory recovery, and on the mechanisms of pro-resolving LMs in their effects on microglial phagocytosis of Aβ42 in vitro[18,21].

We investigated the alterations on phospholipid composition to reveal whether treatment with pro-resolving LMs influences the acyl chains of phospholipids. DHA-containing PS species were significantly elevated upon the treatment of the $App^{NL-G-F}$

mice. Exogenous supplementation with DHA or AA has been shown to alter the composition of membrane microdomains, and incorporation of PUFAs into the phospholipids resulted in a more unsaturated membrane profile[71]. Pro-resolving LMs are released locally at nanomolar levels in a time-dependent manner and are rapidly eliminated. Similarly, remodelling of membrane lipids is temporary, and the time of sample collection and analysis can impact the levels measured. Analysis of endogenous LMs in the present study did not reveal an effect by the treatment of $App^{NL-G-F}$ mice with proresolving LMs. This and the lack of statistical significance regarding endogenous cytokine levels, as well as regarding Aβ pathology, may indicate that the in vivo effects of the treatment are mediated through an indirect route from the periphery. However, the intranasal route is known for its ability to channel substances to the brain and the detection of deuterium-labelled LMs in the brain support that the effects observed in the present study were direct. Nevertheless, mediation of beneficial effects into the brain from the periphery should not be disregarded as an important component of a treatment effect. Differential ability of substances of different types to utilize the intranasal route to the brain is also an obvious factor, and optimization of treatment conditions to fully utilize the central as well as peripheral beneficial effects is a vital part of the development of the treatment strategy presented herein.

In conclusion, we report that intranasal delivery is a non-invasive administration route for LMs that has an impact on the central nervous system. Consequently, we uncover beneficial effects of pro-resolving LMs by improving cognitive functions, rescuing disrupted gamma activity and reduced microglial activation in the $App^{NL-G-F}$ model. Previous studies on LM treatments in AD models have used transgenic models, or models that are peripheral to AD such as the Ts65Dn model. Here, we show that pro-resolving treatment is effective in a model that more resembles a natural "mouse AD" compared to transgenic models where a foreign gene is overexpressed to artificially high levels. We also show that intranasal delivery of pro-resolving LMs is effective in the context of AD. Pro-resolving LMs also altered membrane phospholipid composition by increasing DHA acyl groups in PS. Thus, this study supports the potential therapeutic intranasal delivery of LMs in AD, other neurodegenerative diseases, and various forms of brain injury.

## Methods

**Antibodies and reagents**. The primary and secondary antibodies used are listed in Table 1. LMs were purchased from Cayman Chemicals (USA).

**Animals**. C57BL/6 J wild-type (WT) mice were purchased from The Jackson Laboratory (Bar Harbor, ME, USA). The $App^{NL-G-F/NL-G-F}$ mice[32] were developed at RIKEN Center for Brain Science (Tokyo, Japan) and bred in the Karolinska Institutet animal facility. The $App^{NL-G-F/NL-G-F}$ mice is a model of AD in which the Aβ sequence within the mouse Aβ precursor protein (APP) has been humanized by changes of 3 amino acids and the Swedish, Arctic and Beyreuther familial AD (FAD) mutations were introduced using a knock-in strategy leading to high levels of AD-associated Aβ42[32]. All experiments were performed on male mice housed under pathogen-free conditions in temperature-controlled environment with a standard 12-h light/12-h dark cycle and *ad lib* access to food and water. All handling and experimental procedures were performed in accordance with guidelines for the Comparative Medicine (KM-B, Karolinska Institutet, Sweden). The animal work in this study was approved by the Stockholm Ethical Committee for animal experiments (6-14, 1433-2018, 12370–2019).

**Treatment**. In total, 45 mice at the age of 22–24 weeks were randomly selected and equally divided into three groups. Two weeks before the experiment, the mice were handled every second day. The mice were anaesthetized by 2% isoflurane *via* inhalation in a plastic induction chamber to induce unconsciousness for a minimal duration of time sufficient to complete the administration of proresolving LMs. Intranasal delivery was performed while holding the mouse in a supine position, after which it was kept in the same position for an additional 10-15 s to ensure that the fluid was inhaled. A total of 10 µl of a solution containing non-esterified forms of the LMs RvD1 (CAS No. 872993-05-0), RvD2 (CAS No. 810668-37-2), RvE1

**Table 1 Primary and secondary antibodies and reagents.**

| Protein targeted | Host | Dilution | Provider | Catalogue and lot number |
|---|---|---|---|---|
| **Primary antibodies** | | | | |
| BLT1 | Rabbit | 1:400 | Cayman | 120114; 0495946-1 |
| ChemR23 | Mouse | 1:500 | Santa Cruz | SC-398769; J1216 |
| FPR2 | Rabbit | 1:500 | Santa Cruz | Sc-66901, C2311 (M-73) |
| GABA$_A$R1$\alpha$ | Rabbit | 1:500 | Abcam | Ab252430 |
| GFAP | Rabbit | 1:800 | Dako | Z033401-2; 55769 |
| GLUR1 | Rabbit | 1:400 | Abcam | Ab31232; GR3379169-1 |
| GLUR4 | Rabbit | 1:400 | Merck | AB1508; 3551424 |
| GPR18 | Rabbit | 1:400 | Sigma-Aldrich | SAB4501252 |
| Iba1 | Rabbit | 1:400 | Wako/Fujifilm | 019-19741; CAK 1997 |
| LGR6 | Rabbit | 1:500 | Invitrogen | PA5-102099; VA2931943 |
| PSD95 | Mouse | 1:500 | Abcam | Ab2723; GR3248435-12 |
| Trem-2 | Sheep | 1:400 | R&D Systems | AF1729; JPN0219101 |
| A$\beta$(1-16) peptide | Mouse | 1:500 | BioLegend | 803001; B291304 |
| **Secondary antibodies and reagents** | | | | |
| Alexa Fluor Plus 594 anti-mouse | Donkey | 1:400 | Invitrogen | A32744 |
| Alexa Fluor Plus 488 anti-rabbit | Donkey | 1:400 | Invitrogen | A32790 |
| IRDye®800CW anti-rabbit | Donkey | 1:15 000 | LiCor | 926-32213 |
| IRDye®680RD anti-mouse | Donkey | 1:15 000 | LiCor | 926-68072 |
| IRDye®800CW anti-goat | Donkey | 1:15 000 | LiCor | 926-32214 |
| Intercept® TBS Blocking buffer | | | LiCor | 927-66003 |

(CAS No. 552830-51-0), MaR1 (CAS No. 1268720-28-0) and NPD1 (CAS No. 660430-03-5) at 40 ng per LM (Cayman Chemicals, USA) were administered into the nostrils by a 10 µl pipette to the $App^{NL-G-F}$ mice ($App^{NL-G-F}$-LM) ($n = 13$). Vehicle (0.9% saline) was administered to $App^{NL-G-F}$ mice ($App^{NL-G-F}$-Veh) ($n = 15$) and WT mice (WT-Veh) ($n = 15$). Two $App^{NL-G-F}$ mice were treated with a mixture of deuterium-labelled LMs (RvD1-d$_5$, RvD2-d$_5$ RvE1-d$_4$ and MaR1-d$_5$ (Cayman Chemicals, USA)) ($App^{NL-G-F}$-LM-H). The treatment lasted 9 weeks with administration 3 times a week (Fig. 1a). This protocol was based on the short half-life of the pro-resolving LMs, e.g. around 5 h in plasma for RvD1[72].

Behavioural tests were performed on all mice ($n = 45$). Five (5) mice per experimental group were utilized for biochemical and immunohistochemical analysis, 5 mice per group were used for MALDI-imaging and mass spectrometry analysis, and 5 mice per group for analysis of oscillatory gamma-band activity.

After 7 weeks of treatment, the mice were subjected to behavioural tests. In the 9$^{th}$ week of treatment, the mice were anaesthetized and perfused intracardially with saline, after which organs were dissected out (Fig. 1a). The brains were divided into two hemispheres. The left hemispheres were further dissected into olfactory bulb, hippocampus, cerebral cortex, and cerebellum, frozen in dry ice and stored at −80 °C for biochemistry and liquid chromatography tandem-mass spectrometry (LC-MS/MS) analysis. The right hemispheres were fixed in 4% paraformaldehyde (PF) in 0.1 M phosphate buffer and stored at +4 °C for immunohistochemical studies or frozen in dry ice and stored at −80 °C for MALDI-imaging.

### Behaviour

*Open field.* Exploration and anxiety-like behaviour were analysed using open field (OF) test. The test arena was constructed of a clear plastic (45 cm×45 cm, 40 cm height) (Actimot system, TSE Systems GmbH, Bad-Homburg, Germany) with a video camera placed above the chamber to monitor rearing activity, the distance travelled, and time spent in the central and peripheral area of the arena. Each mouse was placed gently in the centre of the arena under dim light and allowed to explore for 20 min. Before and after each trial, the chamber was cleaned with 70% ethanol and distilled water.

*Elevated plus maze.* The elevated plus maze (EPM) was used to assess differential exploratory tendencies and anxiety-like behaviour in open and closed arms. The maze consisted of four arms that were 35 cm long, 8 cm wide and elevated 50 cm off the floor. The two closed arms had 15 cm high walls and two arms were open. At the beginning of the 5-min session, the mice were placed gently at the centre of the maze and allowed to explore freely. Using Ethovision XT (Noldus) software the number of entries and the amount of time spent in open and closed arms, respectively, were recorded. The time spent in open and closed arms was calculated as the percent of the total time spent in the arms, excluding time in the central area. The maze was cleaned with 70% ethanol and distilled water between trials.

*Novel object recognition.* Novel object recognition (NOR) test is based on the spontaneous tendency for exploring a new object, reflecting learning and recognition memory. The NOR test was conducted with an open field chamber (45 cm×45 cm, 40 cm height). The task consisted of three phases: habituation, familiarization, and test. In the habituation phase (day 1), the mice were allowed to explore freely for 10 min in the open field arena in the absence of objects. During the familiarization phase (day 2), the mice were placed for 10 min in the same open field arena containing two identical objects and then returned to their home cage for 24 h. On the following (test) day (day 3), the mice were placed in the open field arena for 10 min containing two objects, one of which was the same object as during the familiarization (familiar object) and the other was a new object (novel object). Using Ethovision XT (Noldus) software, the discrimination index (DI) was calculated as: (time spent exploring novel object - time spent exploring familiar object) / (time spent exploring novel object + time spent exploring familiar object). DI ranged from -1 to +1, representing exploration of the familiar object with -1, exploration of the novel object with +1, and no preference for objects with 0. The objects in this task were different in shape and colour but similar in size. They were fixed to the bottom of the chamber to prevent movement and the entire set-up was cleaned thoroughly with 70% ethanol and distilled water to remove olfactory cues.

*Fear conditioning.* Contextual and cued fear conditioning (FC) test was used to assess learning and memory with environmental cues and aversive experiences. The mice were placed into a conditioning chamber (black box, 20 cm × 20 cm) to explore freely for 2 min, and then subjected to a 30 s sound stimulus (55 dB, 5000 Hz) followed by a 2-sec electric shock (0.3 mA). The sound and shock were repeated 3 times to strengthen the association (day 1). After 24 h, the mice were returned to the same conditioning chamber for the context test (day 2). Freezing behaviour was recorded for 180 s to analyse contextually conditioned fear. In between the sessions, the chamber was cleaned with 70% ethanol and distilled water on both day 1 and day 2. After another 24 h a cued test (day 3) was conducted. This was performed in a different chamber (round, clear, 20 cm diameter) to provide a new context that was unrelated to the conditioning chamber. The mice were allowed to explore freely for 2 min in the new chamber followed by 2 min with a continuous sound stimulus (55 dB, 5000 Hz). After each mouse, the round chamber was wiped with hypochlorous water (50% diluted) to provide a different environment with a different odour. The tests were recorded with a camera and infrared sensors to detect the animal's location in 3 dimensions using a Multi Conditioning System (TSE Systems GmbH, Bad-Homburg, Germany).

**Electrophysiology recordings and analysis**. Recording of electrophysiology was performed on 10 animals (4 WT-Veh, 3 $App^{NL-G-F}$-Veh and 3 $App^{NL-G-F}$-LMs) after the completion of the behavioural experiments. All chemical compounds used for the intracellular and extracellular solutions were obtained from Sigma-Aldrich Sweden AB (Stockholm, Sweden). Kainic acid (KA) was obtained from Tocris Bioscience (Bristol, UK).

*Hippocampal slice preparation.* Two animals per day were deeply anesthetized using isoflurane before being sacrificed by decapitation. The brains were dissected and placed in ice-cold artificial cerebrospinal fluid (ACSF) modified for dissection containing (in mM): 80 NaCl, 24 NaHCO$_3$, 25 glucose, 1.25 NaH$_2$PO$_4$, 1 ascorbic acid, 3 Na-pyruvate, 2.5 KCl, 4 MgCl$_2$, 0.5 CaCl$_2$ and 75 sucrose. The solution was bubbled with carbogen (95% O$_2$ and 5% CO$_2$). Horizontal sections (350 µm thick) of the ventral hippocampus of both hemispheres were prepared with a Leica VT1200S vibratome (Leica Microsystems). Immediately after cutting, the slices

were transferred to a humidified interface holding chamber containing standard ACSF (in mM): 124 NaCl, 30 NaHCO₃, 10 glucose, 1.25 NaH₂PO₄, 3.5 KCl, 1.5 MgCl₂ and 1.5 CaCl₂, continuously supplied with humidified carbogen. The chamber was held at +34 °C during slicing and subsequently allowed to cool down to room temperature (RT) (~+22 °C) for a minimum of 1 h.

*Electrophysiology.* Recordings were carried out in the hippocampal area *cornu Ammonis* (CA) 3 with borosilicate glass microelectrodes pulled to a resistance of 3-7 MΩ. Local field potentials (LFPs) were recorded using microelectrodes filled with ACSF placed in the CA3 *stratum pyramidale*. LFP oscillations were elicited by applying 100 nM KA to the extracellular bath. LFP recordings and patch-clamp recordings were performed in a Multiclamp 700B (Molecular Devices, CA, USA). In order to maintain stable LFP oscillations all recordings were performed at +34 °C with a perfusion rate of 3–5 ml/min of aerated ACSF containing 100 nM KA. The oscillations were allowed to stabilize for at least 20 min before recording.

Patch-clamp (whole-cell) recordings were performed in FSNs based on their location and their unique electrophysiological characteristics[25]. Action potential (AP) and EPSC measurements (Vh = −70 mV) were performed using a potassium-based intracellular solution (in mM): 122.5 K-gluconate, 8 KCl, 4 Na₂-ATP, 0.3 Na₂-GTP, 10 HEPES, 0.2 EGTA, 2 MgCl₂, 10 Na₂-phosphocreatine, set to pH 7.2-7.3 with KOH, osmolarity 270-280 mOsm. The signals were sampled at 10 kHz, conditioned using a Hum Bug 50 Hz noise eliminator (Quest Scientific, North Vancouver, BC, Canada), software low pass filtered at 1 kHz, digitized, and stored using a Digidata 1440 A and pCLAMP 10.4 software (Molecular Devices, CA, USA).

*Data analysis.* Power spectra density plots (from 60 s long LFP recordings) were calculated in averaged Fourier-segments of 8192 points using Axograph X (Kagi, Berkeley, CA, USA). Gamma oscillation power was calculated by integrating the power spectral density between 20 and 80 Hz. EPSCs were detected offline using MiniAnalysis software (Synaptosoft, Decatur, GA, USA). Charge transfer, event amplitude and interevent-interval were analysed using Microsoft Excel (Microsoft Office) and GraphPad Prism 9 (GraphPad Software, USA), with the result representing average values taken over 1 min periods.

Spike phase-coupling analysis was performed on concomitant LFP recordings and single-unit recordings using MATLAB custom-written routines to relate the FSN spiking activity to ongoing gamma oscillations[25]. To do this, LFP recordings were pre-processed using a band pass filter set to 20-60 Hz (RC-single pole) using Clampfit 10.7. APs were detected using an amplitude threshold and the instantaneous phase of gamma oscillations was calculated using a Hilbert transform to determine the phase-angle at which each AP occurred during ongoing oscillations. Phase-angles and gamma oscillations-phases were represented in polar plots and expressed in radians with the peak of the oscillation cycle corresponding to 0 and the valley corresponding to ±π in the polar plots. To check the synchronization level of the AP firing, the resultant average vector was used to calculate the phase-angle of the APs. This average vector represents the synchronization level of AP firing, and ranges between 0 and 1, where 0 represents firing distribution uniformly throughout the oscillation cycle, while 1 represents firing in a specific phase angle consistently[73]. When all the vectors were assigned, an averaged resultant phase-density vector was calculated to describe the preferred phase-of-firing (phase-angle) and how recurrent the firing at that angle was (vector length). A longer vector length denotes more synchronized AP firing. The vector length is shown normalized by the total number of APs for each condition per cell. The preferred phase-angle was determined cell by cell and calculated by averaging the AP phase-angles at which each cell fired for each condition. To test whether neurons fired in a phase-related manner, all concomitant recordings were tested for circular uniformity using Rayleigh's test. Only recording with *p* values below 0.05 were considered for the analysis.

**Immunohistochemistry.** The tissues placed in 4% PF after dissection (see Treatment above) were kept at +4 °C overnight and then soaked in 30% sucrose for 24 h, after which they were transferred to cryoprotectant solution and stored at −20 °C. The tissues were cut in the coronal plane on a Leica cryostat to obtain 20 µm thick sections. The sections were collected into 24-well plates containing 0.01 M phosphate-buffered saline (PBS) and after three washes the free-floating sections were incubated for 20 min with PBS containing 0.3% Triton X-100 for 15 min at RT. After another wash in PBS, the sections were blocked in a solution containing 5% normal donkey or goat serum and 0.1% Triton X-100 in PBS for 30 min. Subsequently, the sections were incubated overnight with primary antibodies at 4 °C. Normal donkey or goat serum in the absence of primary antibodies was used as a negative control. The sections were washed in PBS and incubated with secondary antibodies for 1 h at RT. The BLT1 and ChemR23 antibodies were validated by pre-adsorption with the immunogen and subsequent finding of absence of signal, and the GPR18 antibodies were validated by the company (https://www.sigmaaldrich.com/SE/en/product/sigma/sab4501253). All of the antibodies resulted in bands at the appropriate molecular weight, and these were used for the analysis.

For Thioflavin-S staining, some sections were incubated in a solution of 1% Thioflavin-S for 8 min and then soaked for 3 min each in 80% and 90% EtOH, and MiliQ, followed by mounting on glass slides and dehydration.

All slides were mounted with Fluoro-Shield (Sigma) mounting medium. Fluorescence images were captured using Leica epifluorescence microscope.

**Image analysis.** The immunoreactivity for ionized calcium-binding adaptor protein 1 (Iba1) and glial acidic fibrillary protein (GFAP) was examined under a Nikon Eclipse E800 microscope (Bergman-Labora, Stockholm, Sweden). Images of cortex (2 fields/section) and hippocampus (1 field/section) were acquired with a 4X objective in a Nikon DS-Qi2 camera with NIS-D 4.3 software (Bergman Labora, Stockholm, Sweden). From each animal, 5 consecutive coronal sections 100–120 µm apart were analysed, in total 15 fields were investigated per animal (*n* = 5). The area occupied by Iba1- or GFAP-positive cells was measured by applying threshold with NIH Image J (United States National Institutes of Health). All images were collected under the same lighting conditions and settings.

Aβ pathology was visualized by immunohistochemistry using antibodies raised against Aβ peptide (6E10). Diffuse and neuritic plaques were analysed in the cerebral cortex and hippocampus areas by analysing composite images of sections double-stained with Thioflavin-S. Images were acquired with a 4X objective in the cortex (2 fields/section) and hippocampus (1 field/section). The number of plaques per area was assessed in 5 consecutive coronal sections 100–120 µm apart from each animal, in total 15 fields per animal (*n* = 5), using NIH ImageJ (United States National Institutes of Health) with the cell counter plug-in. The number of plaques per field was normalized to the area of the field.

**Protein extraction.** Soluble and insoluble protein fractions were obtained from the cerebral cortex and hippocampus. Fresh-frozen brain samples were mechanically homogenised using a 3-step extraction protocol. The tissues were homogenised in TRIS buffer (20 mM Tris-HCl, 150 mM NaCl, pH 7.4) containing cocktails of protease and phosphatase inhibitors (Chemical Co., Stockholm, Sweden, and Thermo Fisher Scientific, Stockholm, Sweden, respectively). After centrifugation (16 000 rpm, +4 °C, 20 min), the supernatant was collected as Fraction 1. The pellet was homogenised in TRIS buffer containing 1% Triton X-100 and centrifuged (16 000 rpm, +4 °C, 20 min). The supernatant was collected as Fraction 2. The remaining pellet was homogenized in 70% formic acid and centrifuged (16 000 rpm, +4 °C, 20 min), and the resulting supernatant collected as Fraction 3.

**Western blot.** The protein concentration in Fraction 1 and 2 was determined by a bicinchoninic acid (BCA) protein assay (Thermo Fisher Scientific). Equal amounts of protein were separated by electrophoresis on NuPAGE 4–12% 15-well Bis-Tris gels (Invitrogen; NP0323BOX) for 1 h at 160 V, and transferred to nitrocellulose membranes (Bio-Rad, USA). The membranes were blocked with Odyssey Blocking Buffer (TBS) (LI-COR 927-50000). The primary antibodies (Table 1) were diluted with their respective blocking buffer and applied to the membranes at +4 °C overnight. After washing with 0.01 M Tris-buffered saline (TBS) with 0.1% Tween-20 (TBS-T) the membranes were incubated with the corresponding secondary antibodies for 1 h at RT and then washed with TBS-T. The blots were scanned in the Odyssey Infrared Imaging System (Li-COR Biosciences) and analysed by performing densitometric analysis using the Image Studio Ver 5.2. All bands were normalized to the corresponding total protein level, detected with Revert Total Protein Stain (LI-COR 926-11010).

**Meso Scale V-plex assays.** Pro- and anti-inflammatory proteins were analysed in Fraction 1 and 2 of the brain homogenates using a 96-well V-PLEX Mouse cytokine 19-plex (Proinflammatory Panel 1 and Cytokine Panel-1) (#K15255D; Meso Scale, Rockville, MD, USA). Each mouse brain extract was diluted two-fold with diluents and incubated for 2 h at RT. After washing with 0.05% Tween-20 in 0.01 M PBS, the plates were incubated with detection antibodies for 2 h at RT.

The levels of Aβ₄₀ and Aβ₄₂ were analysed in Fraction 1, 2 and 3 of the cortex homogenates using a 96-well V-PLEX Aβ₄₂ peptide (4G8) kit (#K150SLE-1; Meso Scale, Rockville, MD, USA). The plate was incubated with recommended diluent for 1 h at RT, washed with 1X MSD Wash Buffer and incubated 2 h at RT with detection antibodies. Data were generated from the V-PLEX Mouse cytokine 19-plex and V-PLEX Aβ₄₂ peptide kits by washing away detection antibody solution and adding reading buffer to generate an electrochemiluminescence signal measured in a Meso Quickplex SQ 120 (Meso Scale, Rockville, MD, USA) and quantified with the Discovery Workbench 4.0 software.

**Lipid extraction and analysis.** Brain tissues were homogenized with CHCl₃/MeOH (2:1) and an internal standard mixture of deuterium-labelled lipids (AA-d8 (5 ng/µl), PGD2-d4 (1 ng/µl), EPA-d5 (1 ng/µl), 15-HETE-d8 (1 ng/µl), and LTB₄-d4 (1 ng/µl)) was added to each sample before sonication for 30 min and storage at −80 °C overnight. The following day, the samples were centrifuged at 4 200 RCF for 30 min and the supernatants collected. The pellets were washed with CHCl₃/MeOH and centrifuged, and the supernatants from both centrifugations were combined. Two ml of distilled H₂O, pH 3.5, was added to each supernatant and after vortexing and centrifugation the pH of the upper phase was adjusted to 3.5-4.0 with 0.1 N HCl. The lower phase was dried down under N₂ and then resuspended in 1 ml of MeOH. Liquid chromatography tandem-mass spectrometry (LC-MS/MS) analysis was performed using a Xevo TQ UPLC (Waters, Milford, MA, USA).

Analysis of the phospholipids phosphatidylcholine (PC), phosphatidylethanolamine (PE) and phosphatidylinositol (PI) was performed in samples dried under $N_2$ and resuspended in 20 µl of the sample solvent ($CH_3CN$/$CHCl_3$/MeOH). The amount for each phospholipid species was calculated as % of the total amount in each sample.

Analysis of FAs and their derivatives was performed in samples dried under $N_2$ and resuspended in 1 ml MeOH. After mixing with 9 ml of $H_2O$ at pH 3.5, the samples were loaded onto C18 columns (Agilent, Santa Clara, CA, USA), and then eluted with methyl formate, dried under $N_2$, resuspended in 50 µl MeOH/$H_2O$ (1:1), and injected into a column. Lipid standards (Cayman, Ann Arbor, MI, USA) were used for tuning and optimization, as well as to create calibration curves for each compound. LC-MS/MS data were analysed following repeated observed retention times for each compound by an estimated limit of detection obtained from the signal to noise ratio (S/N). Low intensity peaks present in the samples were integrated with a width matching the peak width of the compound measured. The area of the compound of interest was then divided by the area of the noise. Two $App^{NL-G-F}$ mice were treated with a mixture of deuterium-labelled LMs for LC-MS/MS analysis. The cortex and hippocampus from each animal underwent lipid extraction. Retention times for the compounds were deduced from repeated observation of elution time and theoretical shift with the addition of deuterium.

**Statistics and reproducibility**. All statistical analyses were performed using GraphPad Prism 9. The results are expressed as scatter plots with median and in bar graphs as mean ± SEM. $P < 0.05$ was considered statistically significant. Kruskal-Wallis and one-way ANOVA were used to test for group differences, with the Dunn *post hoc* test, or manually with Mann-Whitney U test and Bonferroni correction for multiple comparisons to analyse differences between treatments. For electrophysiology, data were filtered from outliers with the ROUT method and assigned to parametric or nonparametric statistical significance analysis after Shapiro-Wilk normality test. Tests for statistical significance were performed using one-way ANOVA with Tukey's post hoc test and Kruskal-Wallis with the Dunn post hoc test. Figure legends note the type of statistical test, the definition of significance for various *p* values, and the number of biological replicates (*n*) for each experiment. For lipid analyses the compound to internal standard ratios were statistically evaluated using one-way ANOVA with Geisser-Greenhouse correction. Tukey's method was applied for the correction of multiple comparisons, determining significance based on $P < 0.05$. MassLynx 4.2 and TargetLynx XS software were used for viewing data and peak integration. Excel and GraphPad Prism 9 were used for statistical analysis.

## Data availability

Source data underlying figures used in the current study are provided in Supplementary Data 1. All other data are available from the corresponding authors upon reasonable request.

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

## Acknowledgements

We thank Drs Takaomi Saido and Takashi Saito at RIKEN Center for Brain Science for providing *App^{NL-G-F}* mice. The authors are grateful for the support from The Swedish Research Council (2018-02601), The Swedish Alzheimer Foundation, Karolinska Institutet research funds, Åhlén-Stiftelsen, Stiftelsen för Gamla Tjänarinnor, Gun och Bertil Stohnes Stiftelse, Stockholm Community Council, Hållstens forskningsstiftelse, Torsten Söderbergs Stiftelse, and the EENT Foundation of New Orleans. All behavioural studies were performed at the Animal Behaviour Core Facility (ABCF) at Karolinska Institutet.

## Author contributions

M.S. initiated the study, C.E., M.S. and N.G.B. designed the experiments, which were carried out by C.E. Behavioural experiments were performed by C.E. and S.G.A., and S.M. provided input. Morphological and biochemical analysis were performed by C.E. and lipid analysis by K.V.D. and B.J. M.O. performed immunohistochemistry and provided some of the graphs and statistical analysis. M.L.C. performed analysis for deuterium-labelled LMs. Electrophysiology recordings were done by L.E.A.-G. and A.F. provided input. P.N. provided the animals. C.E. wrote the paper with input from M.S., N.G.B., S.M., E.H., M.O., L.E.A.-G., P.N. and A.F. All authors have given approval to the final version of the paper.

## Funding

## Competing interests

The authors declare no competing interests.

## Additional information

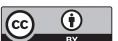

