## [Peer Review File · Communications Biology]

Reviewers' comments:

Reviewer #1 (Remarks to the Author):

In the present work, Emre and colleagues studied the effects of intranasal delivery of SPM in a mouse model of Alzheimer's disease. They show that this SPM mixture attenuated cognitive impairments, restored gamma oscillation deficits, and decreased microglia activation. However, this SPM mixture was unable to alter astrogliosis, amyloid burden or the levels of cytokines and chemokines in the brain. The study has been designed properly, the techniques used are adequate and the results shown are interesting for the scientific and clinical field. I have, however, some concerns

- 1.The authors stated that a total of 200ng of combination of 5 different SPMs (RvD1, RvD2, RvE1, MaR1 and NPD1) was injected per mouse. However, they did not mention whether the proportion of each SPM in this cocktail was homogenous. This should be indicated in the M&M.
- 2.The authors reported that a mixture of deuterium-labelled SPMs (RvD1-d5, RvD2-d5 RvE1-d4 and MaR1-d5) was injected in two mice. I assume this labelled-SPMs were used to investigate whether these SPMs reached the CNS. However, no data is shown about this issue. Did the authors detect the injected SPM in the brain? This is important since they defend the therapeutic exploration of SPM in AD, using a non-invasive route. Why was not NPD1-d5 included in this cocktail?
- 3.The authors stated that the antibodies used in the present study are listed in table 1. However, no tables were included in the manuscript. Please provide this information
- 4.The authors showed that SPMs minimized microglial activation but did not attenuate the levels of pro-inflammatory cytokines or chemokines. However, in the abstract they stated that SPMs decreased microglial activation and proinflammatory cytokines. Please, correct the abstract.
- 5.The authors should speculate (in the discussion) why did SPMs fail to reduce cytokine levels in the brain despite this lipid mediators were able to reduce microgliosis. This is indeed quite surprising, especially, since many reports show that SPMs are able to silence cytokine expression in vitro and in vivo, including in the CNS.
- 6.Line 216: please correct "iquid chromatography"

Reviewer #2 (Remarks to the Author):

Review report COMMSBIO-21-2870-T

In the current study, the authors investigate whether intranasal delivery of pro-resolving lipid mediators could resolve inflammation and ameliorate pathology in an AD mouse model. As endogenous activators of resolution, these LMs gain a lot of attention recently, and have been shown to dampen inflammation in a variety of inflammation-related in vivo models, including AD (Maresin 1, PMID: 31680874). To my surprise, this paper is not mentioned at all, whereas a similar treatment (although in combination with other LMs) was applied in the current study. This should be mentioned as it diminishes the novelty of the study. However, the authors compensate for this by (as said) testing a mixture of LMs (RvE1, RvD1, RvD2, Mar1 The and NPD1) in an AD model, but also by providing more in-depth analysis and exploring a new delivery route, which paves the way to design novel treatment strategies using such a non-invasive route. Overall, this is a well-executed and nicely written study but I do have some concerns that need clarification.

Major concerns

- 1.The authors should mention/discuss the previous study in which maresin 1 was applied in an AD model in all relevant parts of their manuscript.
- 2.Animal model: the pathology in this mouse model starts at 2 months and peaks at 7 months according to the literature. Can the authors explain why they start their treatment roughly after 6 months, and continue that treatment for roughly two months? This also holds true for the behavioral and other tests, were they performed at the right time point considering the above?
- 3.Can the authors provide more details on the used LM's, where methyl ester forms used that are generally more stable in vivo compared to 'normal' LMs? If not, why not?
- 4.Can the authors explain why the currently used treatment protocol was applied (so 3 times a

week). To support that protocol, please provide information on the kinetics of these SPMs in vivo. 5. From a translational point of view, it would have been nice if a control group was included that also received the LMs to see what effect they have on all measured parameters. Can the authors comment on this and explain why this group has been left out?

6. In general, a clear mechanism of action is missing, and I also miss a speculation on this in the discussion section. Many parameters are studied (which is great), but before thinking about a mechanism of action, the first thing we need to know is whether the LMs have a direct or indirect effect. To study that, it would be great if the authors can show that the LMs actually reach the CNS (for example by showing the data as introduced (but not shown?) in the method section: Two AppNL-G-F mice were treated with a mixture of deuterium-labelled LMs. Did the authors find them back in the CNS? In SF3 the authors show endogenous levels of LMs in the CNS, but these are not affected by the treatment, and key cytokines (like IL-10 or TNF α), which are well-known responsive cytokines upon LM treatment, are not different between the treated and non-treated group. These findings at this point together suggest that the tested LMs do not reach the CNS, and therefore might display an indirect role in the observed effects. Please comment on this in the relevant parts of this manuscript (especially the discussion) and provide data (if possible) on this with the deuterium-labelled LMs.

7. In line with that, the authors show WB levels of receptors, but to provide a mechanism of action, it would be better to show the actual IHC to provide spatial information (not for all markers, but maybe a selection for the key receptors based on the working hypothesis on the MoA).

8. Why were no changes observed in A β ? And in line with that, how can the observed in vivo effects be explained? This again is connected to the MoA which needs more attention.

Minor points

1. Rephrase the following sentence in the abstract ('Resolution of inflammation normally follows neutralization of pathogens; and active response to limit damage and promote healing, mediated by pro-resolving lipid mediators (LMs)') which in its current form is difficult to follow. I think the authors here would like to make the connection between chronic inflammation and impaired resolution so please adjust.
2. Please mention in the abstract which specialized pro-resolving lipid mediators were tested and why a mixture was chosen.
3. Introduction section: RvD1 also binds to GPR32, please adjust.
4. Results: the authors sometimes use bar graphs (with error bars), sometimes graphs with individual data points, and sometime a combination of these 2. Please use a similar approach throughout the paper (for example like the figures presented in figure 3).
5. Trem2 western blots are difficult to interpret as multiple (faint) bands appear. Which bands are the correct ones? Can the authors provide better images?
6. The text contains a couple of typos (for example line 216 liquid should be liquid), please adjust and check the text thoroughly.

Responses to reviewers

We are grateful to the reviewers for insightful comments, which have helped us to greatly improve the manuscript. In the following we addressed the issues brought up and describe the changes made (marked in yellow in the manuscript). We hope that by these responses and alterations in the manuscript, the study will now be acceptable for publication in *Communications Biology*. All changes are highlighted in the new version of the manuscript.

Reviewer #1:

1. The authors stated that a total of 200 ng of combination of 5 different SPMs (RvD1, RvD2, RvE1, MaR1 and NPD1) was injected per mouse. However, they did not mention whether the proportion of each SPM in this cocktail was homogenous. This should be indicated in the M&M.

Response: Thank you for pointing out this unclear statement. The animals were given equal parts of the SPMs *i.e.* 40 ng/SPM during 9 weeks with 3 injections per week. It is now clarified in the M&M.

2. The authors reported that a mixture of deuterium-labelled SPMs (RvD1-d5, RvD2-d5 RvE1-d4 and MaR1-d5) was injected in two mice. I assume this labelled-SPMs were used to investigate whether these SPMs reached the CNS. However, no data is shown about this issue. Did the authors detect the injected SPM in the brain? This is important since they defend the therapeutic exploration of SPM in AD, using a non-invasive route. Why was not NPD1-d5 included in this cocktail?

Response: The purpose of the deuterium-labelled SPMs was indeed to be able to trace the SPMs in the brain. We did detect small amounts of the SPMs (Supplementary Fig. S4). In future studies we will establish a time course of the distribution of deuterated SPMs in different brain regions. Regarding NPD1, NPD1-d5 was not commercially available.

3. The authors stated that the antibodies used in the present study are listed in table 1. However, no tables were included in the manuscript. Please provide this information.

Response: Thank you for the information. Table 1 is now included.

4. The authors showed that SPMs minimized microglial activation but did not attenuate the levels of pro-inflammatory cytokines or chemokines. However, in the abstract they stated that SPMs decreased microglial activation and proinflammatory cytokines. Please, correct the abstract.

Response: The abstract has been corrected.

5. The authors should speculate (in the discussion) why did SPMs fail to reduce cytokine levels in the brain despite this lipid mediators were able to reduce microgliosis. This is indeed quite surprising, especially, since many reports show that SPMs are able to silence cytokine expression in vitro and in vivo, including in the CNS.

Response: This was indeed a surprising finding. The effects on microgliosis were detected as reduction in Iba1, a protein involved in microglia movement, and the observed reduction may therefore in part be an effect on microglial movement. However, this is probably not the entire explanation since the morphology of the microglia would suggest an overall activation. The data on cytokines indeed reveal significantly higher levels in the *App^{NL-G-F}* mice than in WT animals, whereas the levels upon treatment with SPMs are somewhere in between for several cytokines. We offer as a consideration that the effects of these LMs may elicit functional

redundancy to mediate resolution of inflammation and additional neuroprotective/neuroactive signalling, as described for NPD1.

Increasing the statistical power by increased number of animals could have given a statistically significant reduction of cytokine levels. The Discussion has been extended in this regard.

6. *Line 216: please correct “iquid chromatography”.*

Response: This has been corrected.

Reviewer #2:

1. *The authors should mention/discuss the previous study in which maresin 1 was applied in an AD model in all relevant parts of their manuscript.*

Response: This is now cited in all relevant parts of the manuscript.

2. *Animal model: the pathology in this mouse model starts at 2 months and peaks at 7 months according to the literature. Can the authors explain why they start their treatment roughly after 6 months, and continue that treatment for roughly two months? This also holds true for the behavioral and other tests, were they performed at the right time point considering the above?*

Response: The treatment was started at 6 months since some several behavioural tests show that impairment starts at this age. There are inconsistencies in the literature as to the appearance of different behavioural changes, and the length of the treatment was chosen to ascertain that there would be detectable impairment compared to WT animals, and that could be altered by the treatment. In addition, the treatment was prolonged since the behavioural testing took 2 weeks, and then there was an extra week for electrophysiology analysis. Furthermore, the translational relevance for the human clinical situation of a treatment starting at an age when behavioural impairment is apparent exceeds the relevance of treatments starting before these occur.

3. *Can the authors provide more details on the used LM's, where methyl ester forms used that are generally more stable in vivo compared to 'normal' LMs? If not, why not?*

Response: The endogenous forms of LMs were used, not the methyl ester forms, and this has now been clarified in the M & M, with addition of catalogue numbers from Cayman Chemicals.

4. *Can the authors explain why the currently used treatment protocol was applied (so 3 times a week). To support that protocol, please provide information on the kinetics of these SPMs in vivo.*

Response: The half-life of SPMs is short, e.g. around 5 hours in plasma for RvD1 as shown e.g. by Yellepeddi et al 2021 (Clin Transl Sci 14:683-691, PMID 33202089), and in order to counteract this we decided to give multiple doses. This has been added to the M & M. It would certainly be better in the future to use more stable forms of the SPMs to reduce the number of administrations.

5. *From a translational point of view, it would have been nice if a control group was included that also received the LMs to see what effect they have on all measured parameters. Can the authors comment on this and explain why this group has been left out?*

Response: This is a very important point and the rationale was simply to optimize the contrast between treated and non-treated animals with pathology, while at the same time limit the

number of animals used. It is indeed very valuable to analyse the effects on WT animals, or e.g. the so called APP-WT, that have humanized APP, but not the 3 FAD mutations introduced in the mouse APP in the *App^{NL-G-F}* mice. The inclusion of WT mice with the same treatment as the AD model will be important for future planned studies.

6. *In general, a clear mechanism of action is missing, and I also miss a speculation on this in the discussion section. Many parameters are studied (which is great), but before thinking about a mechanism of action, the first thing we need to know is whether the LMs have a direct or indirect effect. To study that, it would be great if the authors can show that the LMs actually reach the CNS (for example by showing the data as introduced (but not shown?) in the method section: Two *App^{NL-G-F}* mice were treated with a mixture of deuterium-labelled LMs. Did the authors find them back in the CNS? In SF3 the authors show endogenous levels of LMs in the CNS, but these are not affected by the treatment, and key cytokines (like IL-10 or TNF α), which are well-known responsive cytokines upon LM treatment, are not different between the treated and non-treated group. These findings at this point together suggest that the tested LMs do not reach the CNS, and therefore might display an indirect role in the observed effects. Please comment on this in the relevant parts of this manuscript (especially the discussion) and provide data (if possible) on this with the deuterium-labelled LMs.*

Response: The rationale for using deuterium-labelled LMs was to define that intranasally delivered LMs under our conditions do reach the brain. As shown in a new figure (Supplementary Fig. S4) we could detect small amounts of the deuterium-labelled LMs by LC-MS/MS. The interpretation is therefore that the LMs did reach the brain. A possible explanation for the lack of effect on key cytokines is that the effects seen on behaviour and electrophysiology are not due to their resolving effects on inflammation in the brain, but a direct effect on neuronal functions. Possible mechanisms of the effects observed have now been discussed in the manuscript.

7. *In line with that, the authors show WB levels of receptors, but to provide a mechanism of action, it would be better to show the actual IHC to provide spatial information (not for all markers, but maybe a selection for the key receptors based on the working hypothesis on the MoA).*

Response: This is a valid point that WB data do not give spatial information that can advise on possible mechanism of action. We have performed IHC with antibodies to ChemR23 and BLT1 and in the materials remaining it was not possible to make a definite statement regarding where the changes observed by WB occurred. However, as in the human brain, the mouse ChemR23 staining is found in both neurons and microglia, as indicated in the micrographs shown below (Fig. 1). We have added text in the Discussion with possible implications for mechanism of action.

Fig. 1. Immunohistochemical micrographs of the cerebral cortex of *App*^{NL-G-F} mice after incubation with antibodies to ChemR23. Arrows indicate labelled pyramidal neurons in the left figure and microglia in the figure to the right.

8. *Why were no changes observed in Aβeta? And in line with that, how can the observed in vivo effects be explained? This again is connected to the MoA which needs more attention.*

Response: There can be several explanations to the lack of effect on Aβeta. One possible explanation is that the effects of the LMs are not mediated via the decrease in Aβeta, but a direct effect on microglia, and on neurons mediating the improvement in memory functions. Further discussion on the mechanisms of the *in vivo* effects is now included in the Discussion.

Albeit not statistically significant, the total levels of 6E-10 labelling in Fig. 3D, show somewhat lower levels in the treated mice. However, it may be difficult for this kind of treatment to affect amyloid plaques, and as mentioned in the Discussion, unfortunately the homogenization procedure used for the biochemical analyses precluded detection of monomeric or oligomeric forms of Aβeta, *i.e.* soluble forms that could conceivably be affected by the treatment.

Minor points:

1. *Rephrase the following sentence in the abstract (‘ Resolution of inflammation normally follows neutralization of pathogens; and active response to limit damage and promote healing, mediated by pro-resolving lipid mediators (LMs)’) which in its current form is difficult to follow. I think the authors here would like to make the connection between chronic inflammation and impaired resolution so please adjust.*

Response: The sentence has been adjusted.

2. *Please mention in the abstract which specialized pro-resolving lipid mediators were tested and why a mixture was chosen.*

Response: This has been added to the abstract.

3. *Introduction section: RvDI also binds to GPR32, please adjust.*

Response: This has been added to the Introduction.

4. *Results: the authors sometimes use bar graphs (with error bars), sometimes graphs with individual data points, and sometime a combination of these 2. Please use a similar approach throughout the paper (for example like the figures presented in figure 3).*

Response: The figures have been altered using a similar approach as in Fig. 3D and E, *i.e.* bar graphs with individual data points for all figures except for the behavioural data in Fig. 1, which are shown as scatter plots, which we believe is more appropriate for these data. In the case of the bar graphs in Fig. 2, we have not added the individual data points in view of the large number of individual data points which make them too busy. A possibility would be to make them as scatter plots, but these data points do not represent animals and thereby misleading. We can of course change them too, should it be requested.

5. *Trem2 western blots are difficult to interpret as multiple (faint) bands appear. Which bands are the correct ones? Can the authors provide better images?*

Response: The MW for Trem-2 is 35 kD. Unfortunately, the Western blots for TREM-2 have this appearance with bands between 20 and 40 kD, similar to our finding in previous studies (Emre et al 2021. Acta Neuropathol. Commun. 9:116).

6. *The text contains a couple of typo's (for example line 216 iquid should be liquid), please adjust and check the text thoroughly.*

Response: This has been adjusted, and we have made an additional check for typo's.

REVIEWERS' COMMENTS:

Reviewer #1 (Remarks to the Author):

The authors have satisfactorily addressed all my concerns

Reviewer #2 (Remarks to the Author):

The authors have adequately addressed all my points and have made subsequent adjustments to all relevant sections in the text. One final (minor) aspect is that the authors should state in line 234/235 that the deuterium-labelled LMs were detected in the brain according to SF4 (now it states that they are only detected, not specified where). And the authors should conclude based on these preliminary findings that SPMs can actually reach the brain where they can exert their functions (such a short conclusion should be added to this part as well, as contains important information for the reader).

Responses to reviewers

We are grateful to the reviewers for their renewed review of our manuscript, and to Reviewer #2 for pointing out the additional changes to be made.

We hope that by doing these changes, the study will now be acceptable for publication in Communications Biology.

Reviewer #2:

One final (minor) aspect is that the authors should state in line 234/235 that the deuterium-labelled LMs were detected in the brain according to SF4 (now it states that they are only detected, not specified where). And the authors should conclude based on these preliminary findings that SPMs can actually reach the brain where they can exert their functions (such a short conclusion should be added to this part as well, as contains important information for the reader).

Response: The following sentence is now added in lines 230-232:

The deuterium-labelled LMs administered in two of the *App*^{NL-G-F} mice were detectable by LC-MS/MS (Supplementary Fig. 5) and support that the intranasally administered LMs reach the brain where they can exert their functions.